

# A 45-year hydrological and planktonic time series in the South Bight of the North Sea

David Devreker[1], Guillaume Wacquet[1], Alain Lefebvre[1]

[1]Ifremer, COAST, 150 Quai Gambetta, F-62321 Boulogne-sur-Mer, France

*Correspondence to*: David Devreker (david.devreker@ifremer.fr)

**Abstract.** This article presents a 45-year data series (from 1978 to 2023) acquired under the IGA ("Impact des Grands Aménagements" in French; Impacts of Major Developments) program, conducted by Ifremer for EDF (Électricité de France, the French multinational electricity utility company). The IGA program was established to monitor the ecological and environmental quality of the coastal area surrounding the Gravelines Nuclear Power Plant (NPP), located in the southern bight

of the North Sea. The main objective of this program is to assess medium- and long-term environmental evolution by providing means to identify possible changes in the local marine habitats. Since 1978, the IGA program has measured key parameters, including temperature, salinity, nutrient concentrations, oxygen levels, chlorophyll-*a* concentrations, as well as the abundance of phytoplankton and zooplankton species. These measurements have been taken at different sampling stations around the NPP, including the "Canal d'amenée" sampling station, for which hydrological and biological characteristics are considered

as representative of broader coastal area of the southern bight of the North Sea. This data paper provides an overview of the main statistical characteristics of the time series, including long-term trends and shifts analysis. Despite the importance and length of this dataset, one of the longest available for this region, its application in advancing knowledge of hydrological and biological processes has been surprisingly limited. The aim of this paper is to make this valuable dataset available to the scientific community, stakeholders, and society to help decipher the local and global influence of anthropogenic activities in a

world increasingly affected by climate change. Since all the main statistics and patterns are still available thanks to our analysis, the user should be able to use this data and combine it with other sources (*in situ*, satellite, modelling), in order to dive into deeper analyses, and to investigate new key scientific challenges as well as more specific ones.

## 1 Introduction

Understanding dynamics of marine ecosystems requires long-term time series that allow to take into account different sources

of natural variability (seasonal, inter-annual) as well as anthropogenic variability (under local/regional pressures, global warming). However, such long-term series are difficult to obtain, as they are costly to maintain and require measurement protocols that are stabilised throughout their duration to avoid acquisition biases. Especially, long-term planktonic time series, particularly for phytoplankton and zooplankton, are relatively scare and not widely available in the scientific literature.



Phytoplankton form the basis of pelagic food chains and contribute to 50% of the Earth's chlorophyll biomass (Falkowski et

al., 2003), playing a crucial role in capturing carbon dioxide ($CO_2$) and producing oxygen ($O_2$). Under certain favourable conditions of temperature, light, turbulence and nutrient availability, the growth of various phytoplankton species can be enhanced. Among the species forming massive blooms in the Eastern Channel/South Bight of the North Sea (SBNS), some, like *Pseudo-Nitzschia*, *Dinophysis*, and *Alexandrium*, are known to produce phycotoxins that pose significant health risks to humans when introduced into the food chain, particularly through filter-feeding shellfish. Other species, such as *Phaeocystis*

*globosa* (Lefebvre and Dezécache, 2020; Karasiewicz and Lefebvre, 2022; Karasiewicz et al., 2018; Lancelot and Rousseau, 1994), can become so abundant that they disrupt ecosystem functioning. Both physico-chemical parameters (bottom-up control) as well as zooplankton populations (top-down control) influence the phenology of phytoplankton species (Banse, 1992, Feng et al., 2014).

Zooplankton is the main aquatic animal compartment in terms of biomass and diversity (Mauchline, 1998). Zooplankton is

considered as primary consumer, as it is a major consumer of phytoplankton (Atkinson, 1996), it makes this organic matter available to predatory animals, particularly fish larvae and juveniles. In the Eastern Channel/SBNS holoplankton (i.e. species that stay planktonic during all their life cycle, by opposition to meroplankton) are dominated by copepod species such as the calanoids *Temora longicornis*, *Acartia clausii* and *Centropages hamatus* and the harpacticoid *Euterpina acutifrons* (in order of annual phenology and dominance, Brylinski, 2009). Zooplankton communities are generally studied on an ad hoc basis, and

long-term zooplankton time series are even rarer in the scientific literature than phytoplankton time series especially in the Eastern Channel and in the French part of the SBNS.

The ecological monitoring of Nuclear Power Plant (NPP) discharges into the sea focuses on studying the medium- and long-term temporal evolution of various marine domains, including pelagic, benthic and fishery domains, along with their associated compartments (hydrology, physico-chemistry, chemistry, phytoplankton, zooplankton, benthos, microbiology, etc.). This

monitoring is conducted on a localized spatial scale, focusing on areas surrounding the power plants and within their zones of influence. The aim is to detect any changes by monitoring specific parameters that are characteristic of each marine compartment. Ecological monitoring of such infrastructure in Gravelines, located in the French coastal part of the Dover Straight, was initiated in 1978 providing the longest-running ecological time series in this part of the English Channel–North Sea continuum. The monitoring strategy and sampling methods (including sampling locations and frequencies, nature of

analyses carried out) evolved between 1978 and 1986 as the six production units of the Gravelines power plant came on stream. One of the monitoring stations (named "Canal d'amenée") of this network has been specifically set up to track the characteristics of seawater entering the cooling turbines and is therefore not influenced by the NPP activity. This monitoring station, crucial for the smooth operation of the NPP, has benefited from an improvement of the monitoring protocol, by increasing the frequency of measurements. Particular attention has been paid to plankton monitoring, where certain species

(gelatinous zooplankton or HABs such as *Phaeocystis globosa*) are known to cause seasonal disruptions to the NPP operations (Masilamoni et al., 2000; Wang et al., 2022; Wang et al., 2023). Initially established to assess the direct and indirect effects of



large-scale coastal developments, this monitoring effort now also offers scientists a valuable opportunity to study local plankton dynamics on a multi-decadal scale.

## 2 Objectives

The aim of this paper is to present the IGA physico-chemical, phytoplankton and zooplankton dataset at the "Canal d'amenée" sampling point. This includes an overview of the sampling strategy, data collection process (with associated Quality Assurance/Quality Controls steps), data investigation, and storage. The characteristics of the different datasets as well as a general interpretation of their variability will be detailed. Based on the limited existing applications and valorisations of the IGA dataset, we will demonstrate its relevance not only for furthering understanding in marine phytoplankton ecology, but

also for public policy needs, such as assessment of environmental or ecological status as requested by EU directives and Regional Sea Conventions. Additionally, we introduce some numerical tools based on an R-package available for the scientific community and developed specifically to rapidly process such data and therefore to valorise the findings.

## 3 Materials and methods

### 3.1   Sampling point location

The Gravelines NPP site is located in the southern bight of the North Sea, near the Dover Strait. This situation, coupled with the shallow waters (less than 30 meters), results in strong hydrodynamic conditions. The hydrodynamic regime of the southern North Sea is influenced by the semi-diurnal tidal circulation, and wind action. The tidal range is macrotidal, varying from 3.5 meters during neap tides to 5.6 meters during spring tides. Tidal currents along the Dunkirk coast run parallel to the coast and are asymmetrical: the flood tide, flowing northeast, is faster and more intense than the ebb tide, which flows southwest. This

flood current dominates the entire southern North Sea coastline (Figure 1), with recorded maximum speeds of approximately 1.5 $m.s^{-1}$. Off Dunkirk, "the flood is present between 3 hours before and 3 hours after high tide" (SHOM, 1988). The only river flowing into the North Sea along the French Coast is the Aa, an 89-kilometer-long river with a watershed of 1,215 km² and a discharge rate of 10 $m^{-3}.s^{-1}$. Industrial activities represent the main local pressures on this environment.

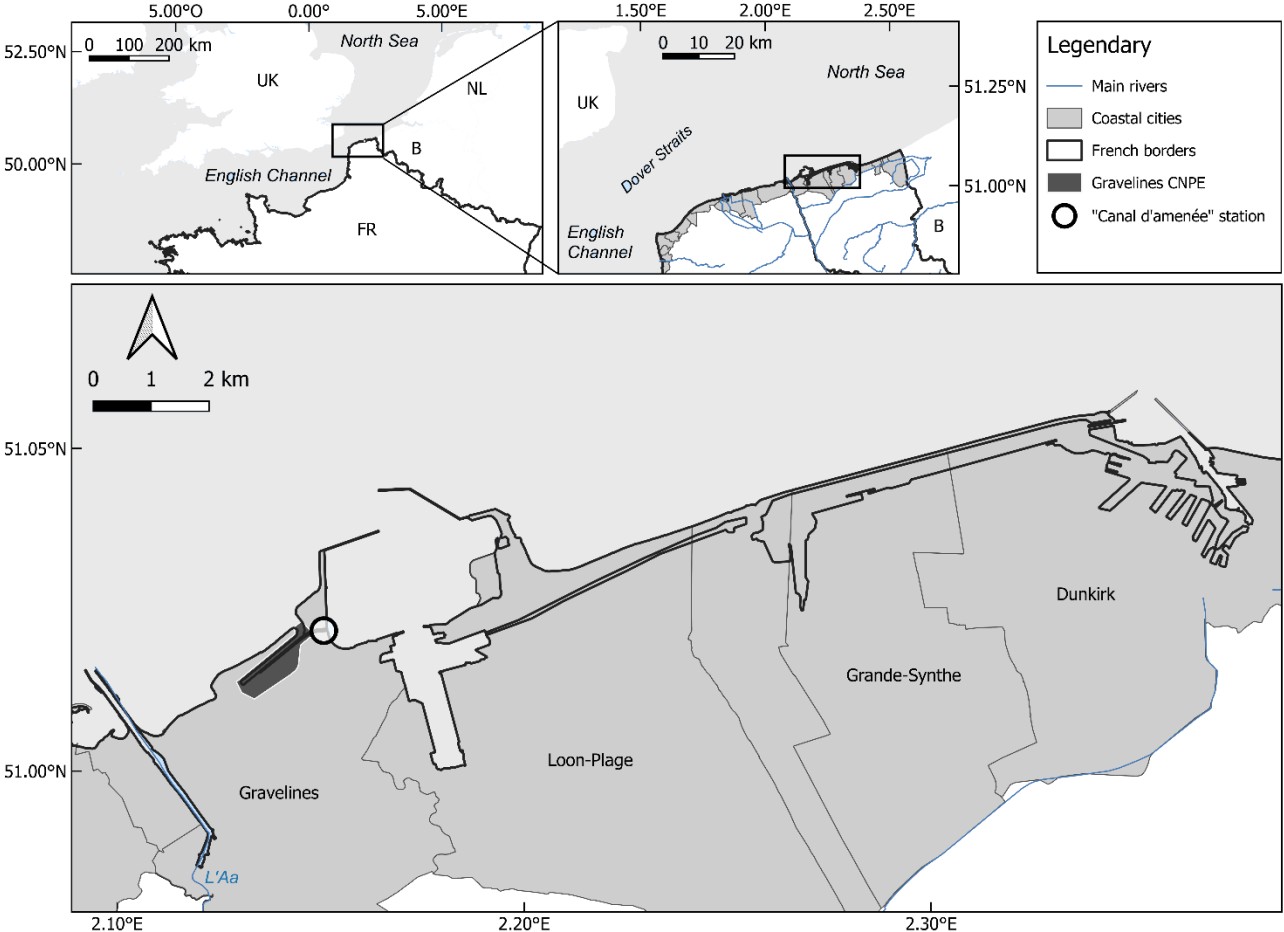

**Figure 1: Location of the "Canal d'amenée" sampling station (black circle) in the Gravelines harbour, which is open on the South West bight of the North Sea.**

The most comprehensive ecological time series are available from the "Canal d'amenée" sampling station (x: 2.15003, y: 51.0222), located within the Gravelines harbour at the entrance of the NPP cooling system (Figure 1). This position allows the station to avoid the direct impacts of the NPP operations, although it remains subject to other anthropogenic influences such as global warming and eutrophication. This is therefore a relevant position for coastal monitoring, completing the monitoring network of the French North Sea coastal ecosystem alongside the SRN Dunkerque 1 station (Lefebvre and Devreker, 2023).

## 3.2 Hydrological/biological parameter measurements

Water samples were initially collected at a monthly frequency and later at a weekly frequency at the end of the 80's, at the surface of the "Canal d'amenée" sampling station using an oceanographic bucket. The water is analysed for different parameter measurements, though not all parameters were measured in every instance. The evolution of the monitoring practices and the



specific temporal distribution of samples are detailed by parameter in the Supplementary Materials (Table S1 and Figures S1 to S3).

### 3.2.1 Temperature, salinity, turbidity and oxygen concentration

These four variables are measured using a multi-parameter probe. Salinity is determined using a conductivity sensor coupled with a temperature sensor (°C) and is expressed as a conductivity ratio (PSU for Practical Salinity Unit). Unlike temperature measurements, which are conformed to the global frequency strategy (i.e. monthly sampling until 1985, then weekly since), salinity was recorded only twice a year between 1990 and 2006, and has been monitored weekly since 2013.

Since 2019, turbidity is assessed by nephelometry and expressed in Nephelometric Turbidity Unit (NTU) (US-EPA, 1980), but was measured with a FNU (Formazine Nephelometric Unit, ISO 7027) sensor between 2016 and 2018. Turbidity is 105 quantified by measuring the amount of light scattered at 90° to the incident light. Dissolved oxygen concentration has been measured using a luminescence oxygen sensor since 2016 and is expressed in mg.l$^{-1}$ (NF EN ISO 25814). However, prior to 1987, it was measured using the Winkler method and reported in ml.l$^{-1}$. The complete methods are described in Aminot and Kérouel (2004). Oxygen concentrations were measured only in the 70's, 80's and since 2016.

Except for turbidity, these variables are recognized as Essential Ocean Variables (EOVs) for physics and biochemistry by the 110 Global Ocean Observing System (GOOS) (Muller-Karger et al., 2018).

### 3.2.2 Nutrients

For nutrient analysis, a 65 ml subsample (for phosphate and Dissolved Inorganic Nitrogen DIN = ammonium + nitrite + nitrate) or a 125 ml subsample (for silicate) of water filtered through a 48 µm mesh is frozen at -25°C. Nutrient concentrations are subsequently measured in the lab using a spectrophotometer with an Optical Density (OD)/concentration relationship 115 determined from a calibration curve performed for each series of measurements. The complete methods are detailed in Aminot and Kérouel (2007).

Ammonium ($NH_4^+$) is quantified using the indophenol blue molecular absorption spectrophotometric method (NF T90-015-2, concentration expressed in µmol.l$^{-1}$). Nitrite ($NO_2^-$) and nitrate ($NO_3^-$) are measured using the N-naphthyl-ethylenediamine 120 molecular absorption spectrophotometric method (NF ISO 13395, concentration expressed in µmol.l$^{-1}$). Phosphate ($PO_4^{2-}$) is determined by the phosphomolybdic blue molecular absorption spectrophotometric method (NF ISO 6878, concentration expressed in µmol.l$^{-1}$). Silicate (SiOH) in water is measured using the silicomolybdic blue molecular absorption spectrophotometric method (NF T90-007, concentration expressed in µmol.l$^{-1}$).

Nutrient monitoring was suspended in the mid-1990s and in the 2000s for nitrates and nitrites. Phosphate measurements were 125 more intermittent, conducted only from 1978 to 1986 and then resuming from 2016 to the present at a monthly frequency. The Global Ocean Observing System also classifies nutrients as Essential Ocean Variables for biochemistry.





### 3.2.3 Chlorophyll-*a*

To measure chlorophyll pigments, an indicator of phytoplankton biomass, one liter of water is kept cool and protected from light. For determining the specific composition and abundance of phytoplankton, 500 ml of water is fixed with Lugol's solution

(2.5 ml per liter of seawater). Until 2018, chlorophyll-*a* concentrations were determined using the trichromatic method (SCOR-UNESCO, 1966). From December 31, 2018, the measurement method used is the monochromatic method (Lorenzen, 1967). Regardless of the method, one-liter water samples are filtered through Whatman GF/C 47 mm glass fiber filters. The chlorophyll pigments concentrated on these filters are then extracted with 90% acetone. After centrifugation for 20 minutes at 6,000 rpm, the absorbance of the supernatant is measured spectrophotometrically. The limit of quantification is 0.10 µg.l⁻¹.

Chlorophyll-*a* concentration (proxy of phytoplankton biomass) is measured according to the global frequency strategy and is recognized as a Biology and Ecosystems EOV by the GOOS.

### 3.3  Plankton

### 3.3.1 Phytoplankton

A 10 ml volume of the Lugol-fixed water samples is decanted in a sedimentation tank for at least 12 hours, following the

method of Utermöhl (1958). Cell counts are then performed using inverted microscopy within a month of sample collection to minimize significant changes in phytoplankton size and abundance. Except for *Phaeocystis globosa* enumeration, over 200 phytoplankton cells per sample are counted using a 20X Plan Ph1 0.5NA objective, yielding an error margin of 10%. For *P. globosa*, only the total number of cells is computed. A minimum of 50 solitary cells are enumerated from several randomly selected fields (10 to 30) using a 40X Plan Ph2 0.75NA objective. The abundance of cells in a colony is determined using a

relationship between colony biovolume and cell number, as defined by Rousseau et al. (1990). No counting data for *P. globosa* is available between 1982 and 1990 due to unknown reasons.

Phytoplankton identification is standardized using the WoRMS (2024) database and reaches the species level in many cases (Table 1). However, when identification is challenging or uncertain, a lower taxonomic level is kept. Some species are also grouped into "artificial taxa" or "Complex" of species or by common higher taxonomic ranks (mainly Genus or Family) if

they are subject to strong identification confusion (this is the case for *Pseudo-nitzschia* or *Chaetoceros*, for example). In such cases, "[]" denotes a complex of species, while "+" indicates a complex of species and common genus. These groupings are considered as taxonomic units to maintain consistency in the database, despite changes in taxonomic names. Freshwater phytoplankton and protozoa, though regularly found in low abundances in samples, are not considered in figures and statistical analyses that only focus on marine species. Species richness calculations, the Shannon-Weaver index (Equation 1), are based

on all taxonomic levels:

$$H' = -\sum_{i=1}^{S} p_i . \log \log p_i \qquad (1)$$



During the period from 1978 to 2023 (excluding 1987 which is a missing year), a total of 1,811 phytoplankton samples were collected, representing 237 taxa across various taxonomic ranks, which are mostly species and genera, providing a good overall taxonomic accuracy (Table 1). Among these taxa, the Prymnesiophyceae species *Phaeocystis globosa* is the most abundant,

with blooms reaching "more than millions" of cells per liter each year. The six other prevalent taxa are Bacillariophyceae species (i.e. *Rhizosolenia delicatula*, *Chaetoceros*, Pse*udo-nitzschia* [*calliantha* + *delicatissima* + *pseudodelicatissima* + *subcurvata*], *Rhizosolenia* sp., *Skeletonema costatum*, *Leptocylindrus* [*danicus* + *curvatus* + *mediterraneus* + *aporus* + *convexus* + *hargravesii* + *adriaticus*]). The seventh most common taxa are Cryptophyceae, while the first Dinophyceae genus is *Gymnodinium*, ranked at the 37th position. However, the relative dominance of these taxa can vary with seasons and years.

Phytoplankton sampling frequencies were conformed to the global frequency strategy, and phytoplankton diversity is recognized as an EOV for Biology and Ecosystems by the GOOS.

**Table 1: Number and occurrence of the different taxonomic ranks (highest level of taxonomic identification) encountered in phytoplankton identification.**

| Taxonomic rank | Number | Occurrence |
|---|---|---|
| Kingdom | 1 | 2 |
| InfraPhylum | 1 | 209 |
| Phylum | 3 | 934 |
| Form | 2 | 4 |
| Class | 6 | 1,243 |
| Order | 5 | 928 |
| Family | 7 | 99 |
| Complex | 6 | 1,273 |
| Genus | 107 | 14,740 |
| Species | 112 | 17,972 |

**3.3.2 Zooplankton**

Zooplankton sampling at the "Canal d'amenée" station was conducted monthly using a WP2 net. Over time, the type of plankton net used has changed. From 1991 to 2007, the standard WP2 net, used since the start of monitoring in 1978, was replaced with a smaller net (0.09 m$^2$ opening, 110 cm high, tapered, 200 µm mesh size) for sampling in the intake channels. The smaller opening and conical shape of this net made it more sensitive to the effects of clogging and backflow, particularly

in spring and summer when algae (*Phaeocystis globosa*, *Coscinodiscus* sp...) or Noctiluca proliferate. The WP2 net, with its wider opening and cylindrical-conical shape, prevents backflow into the net and is more effective for capturing zooplankton from a single point. From 2008 onward, the standard WP2 net was reintroduced for sampling. A 2018 analysis of potential biases related to changes in sampling strategies (frequency, net type, level of identification) indicated that the replacement of the smaller net by a WP2 net did not significantly impact the estimates of total zooplankton abundance.

Samples are filtered through a 200 µm mesh at the lab, and subsamples are obtained using a Motoda Box and identified under a binocular microscope in a Dolfus tank. Identification stops when 100 individuals are counted. The number of individuals counted is then extrapolated to the total sample volume (ind.m$^{-3}$). Zooplankton identification is standardized using the WoRMS



(2024) database, achieving species-level identification in many cases (Table 2). However, as for phytoplankton, when identification is challenging or uncertain, a lower taxonomic level is retained. To prevent misinterpretation of scientific names, the AlphaID, which comes from the WORM reference website, is also provided. Species richness calculations, the Shannon-Weaver index (Equation 1), are based on all taxonomic levels (not only species).

During the 1978–2023 period, 585 zooplankton samples were collected, representing 224 taxa across different taxonomic ranks which are mostly species and genus, providing a high taxonomic accuracy. The most abundant species identified is the Calanoid copepod *Temora longicornis*. Copepods are globally the most abundant zooplankton taxa (*Acartia clausi*, *Euterpina acutifrons*, *Centropages hamatus*, *Pseudocalanus elongatus* and *Paracalanus parvus*). Additionally, the appendicular *Oikopleura dioica* is frequently observed. However, this relative dominance varies among seasons and years. Zooplankton diversity is identified as a Biology and Ecosystems EOV by the GOOS.

**Table 2: Number and occurrence of the different taxonomic ranks (highest level of taxonomic identification) encountered in zooplankton identification.**

| Taxonomic rank | Number | Occurrence |
|---|---|---|
| Kingdom | 1 | 87 |
| Subclass | 2 | 503 |
| Infraorder | 4 | 393 |
| Phylum | 8 | 969 |
| Order | 12 | 1,136 |
| Class | 12 | 1,288 |
| Family | 24 | 760 |
| Genus | 53 | 1,226 |
| Species | 108 | 6,091 |

### 3.4 Other parameters

The IGA survey monitor additional parameters, but these were measured only sporadically over a few years, depending on the monitoring strategy. These parameters, including pheopigments, zooplankton biomass, pH, suspended organic matter, nitrogen and carbon organic concentrations in zooplankton, are scattered across the survey period and have therefore not been included in this article.

### 4   Database

To efficiently manage coastal monitoring data, Ifremer has developed the Quadrige² information system (https://envlit.ifremer.fr/Quadrige-la-base-de-donnees; last access on April12, 2023), which combines a database with various products and services. Quadrige² plays a crucial role in two key areas: (1) securely and optimally storing basic monitoring data, including analysis results from all monitoring networks, in a supervised and scalable manner, and (2) interpreting and enhancing the value of this data. Once data is stored with an assigned quality level, it becomes available for a wide range of applications. As a result, this system is the required link for monitoring data between data collection in the field and its





availability in multiple formats. Quadrige² has been approved as the national reference information system for coastal waters by the French Ministry of the Environment and is part of broader national data portals dedicated to ocean data such as the

ODATIS Pole, which is part of the Research e-Infrastructure Data Terra (https://www.data-terra.org/).

The datasets presented in this article are derived from extractions from the Quadrige² database. The raw extraction, which includes all parameters (hydrological, phytoplankton, and zooplankton) is available on SEANOE. The data is provided as a semicolon-delimited CSV file, but it contains non-ASCII characters and a few errors that have accumulated over time. Consequently, a data pre-processing phase was applied to the extraction, including checks for duplicates, outliers, the accuracy

of identified taxa, and the verification and harmonisation of measurement units. This ensured that the dataset was clean and reliable for further analysis.

Some cells in these databases are empty, particularly in the "hours" column, where the sampling time was not consistently recorded by operators. Additionally, the metadata for the "$NO_3+NO_2$" parameter is only filled when the values result from direct sample analyses, not from the addition of NO2 and NO3 results, which was done for database harmonization.

Consequently, the final quality-controlled database is divided into three separate semicolon-delimited CSV files, all encoded in UTF-8 with ASCII characters and using a dot as the decimal separator. The first file, containing hydro-chemical parameters and chlorophyll-*a* concentrations includes 19 columns and 12,635 rows. The second file, which holds phytoplankton abundance data, has 20 columns and 37,418 rows. The third file, dedicated to zooplankton abundance data, contains 21 columns and 12,453 rows.

The database header columns are in French, as Quadrige² is a French national database. A French-English translation of these headers is provided in the Supplementary Materials, and a detailed description can be found in Ifremer (2017). Physico-chemical, phytoplankton and zooplankton data are available for the 1978-2023 period, because the database is up to date until 2023 at the time the article is written. The datasets will be updated annually in SEANOE, maintaining the same DOI to ensure continuity and accessibility.

## 5 Quality Control

### 5.1 Data Validation

The data are collected in the field and/or laboratory and subsequently entered into the Quadrige² database via a dedicated user interface. Data control involves verifying and potentially modifying the entered data (including both results and metadata) to ensure consistency with the original bench book (or field sheets). After this verification and any necessary corrections, the data

follow this validation process:

- data validity: ensures the accuracy and reliability of the data corresponding to the analytical results,

- data locking: secures the data, preventing further modifications, even by the original data entry person,

- data distribution: once verified, the data become accessible for extraction and dissemination by all authorized Quadrige² users.



## 5.2 Data Qualification

Following the initial round of data verification, the data undergo a qualification procedure which involves:

- searching for data may be scientifically suspicious or clearly erroneous/aberrant,

- correcting data where possible, making adjustments to correct any identified issues,

- assigning a qualification level, which is:

- good: the data is scientifically valid and relevant,

- doubtful: the data may be inaccurate: taking it into account may bias the results,

- false: the data is considered erroneous or problematic, and should not be included in the analysis.

The level of qualification reflects the level of confidence in the data. It determines the way in which the data is distributed (only data qualified as "good" and "doubtful" are widely distributed), and how it is used in specific data processing. It is determined through a two-steps qualification process:

1) "automatic" qualification, that involves identifying obvious and easily detectable errors in the data,

2) "expert" qualification, which focuses on detecting statistically aberrant data using adapted methods (time series analysis, statistical tests...). Only data qualified as "good" or "doubtful" from the previous step are used for this expert qualification.

## 6    Data analysis

The R-package TTAinterfaceTrendAnalysis was used to efficiently, homogeneously and rapidly perform these tests and extract the most relevant statistical metrics (Devreker and Lefebvre, 2014). Temporal trend analysis was performed using the Seasonal Mann-Kendall non parametric test, with p-values corrected for autocorrelated data. Since this test is suited for monotonic trend analyses, the cumulative sum method was applied to identify shifts within the time series. Moreover, when a significant trend was detected, the Theil-Sen slope estimator was used to quantify the magnitude of the trend.

## 7    Data summary

Table 3 presents the descriptive statistics for each physicochemical and biological parameter at the "Canal d'amenée" station. At an inter-annual scale, temperature is the only hydrological parameter showing a significant upward trend between 1978 and 2023, with an average increase of +0.04°C per year, resulting in a total rise of +1.8°C during the survey period. Among nutrients, only phosphorus shows a significant decreasing trend, with a rate of -0.01 µmol.l$^{-1}$.year$^{-1}$. Chlorophyll-*a* concentrations also demonstrate a significant decreasing trend with a clear shift in the time series (Figure 2, Table3 and Figure 4). This shift, detected by the Pettitt test, occurs around 2012, with mean chlorophyll-*a* concentrations decreasing from 5.85 µg.l$^{-1}$ before 2012 to 2.68 µg.l$^{-1}$ afterwards (and from 8 µg.l$^{-1}$ to 3.7 µg.l$^{-1}$ during the growing season from March to September).





At the inter-seasonal scale, nutrient concentrations (nitrate+nitrite, silicate and phosphorus) are maximum in winter (Figure 3). These concentrations decrease in February/March when concentrations in chlorophyll-*a* (Figure 3) and phytoplankton abundances (Figure 5) increase. Chlorophyll-*a* concentrations reach their maximum from March to July, with the highest average value occurring in April (12 µg.l$^{-1}$), and the maximum value reached in March 2010 at >60 µg.l$^{-1}$. Temperature follows a typical pattern for temperate coastal waters, with February being the coldest month (6.9°C) and August the warmest (19.6°C)

(Figure 3). Mean oxygen concentrations are highest in winter (~10 to 11 mg.l$^{-1}$), when photosynthetic activity and temperature are lowest. These levels decrease in spring, reaching 8 to 9 mg.l$^{-1}$ (Figure 3).

Inter-annual variability of phytoplankton communities reveals a significant increase in the abundance of major taxonomic classes including Bacillariophyceae (and its major species), Dinophyceae and Cryptophyceae. The species *Phaeocystis globosa* does not show any significant trend. On an annual scale, phytoplankton communities are typically dominated by either *P.*

*globosa* or Bacillariophyceae, alternating between the two (with Bacillariophyceae dominance observed in 1993-1997, 2002-2005 and 2013) (Figure 6). Mean annual abundances tend to be higher during years where *P. globosa* is dominant (Figure 6). The ratio Dinophyceae/Bacillariophyceae abundance (an indicator of eutrophication status, Wasmund et al., 2017; Xiao et al., 2018) shows a significant increasing trend, suggesting that Dinophyceae abundances are increasing more rapidly than Bacillariophyceae (Table 3).

Phytoplankton communities also show a clear seasonal pattern. Abundances are low in early winter and begin to increase with Bacillariophyceae starting during February. From March to May, *Phaeocystis globosa* is dominant, constituting over 90% of total phytoplankton abundance, while Bacillariophyceae remain dominant for the rest of the year. *Phaeocystis globosa* bloom corresponds to the peak period of phytoplankton abundance and the lowest specific richness (Figure 7). During the average *P. globosa* bloom, Bacillariophyceae species such as *Chaetoceros* or *Asterionellopsis* are present at the beginning of the bloom,

followed later by *Guinardia* and *Pseudo-nitzschia*.

Concerning zooplankton communities, the mean seasonal variability shows dominance of meroplankton in February while copepods are clearly dominant the rest of the year (Figure 8). Zooplankton mean monthly abundance show a pic value in March at 7,200 ind.m$^{-3}$ when the copepod *Temora longicornis* is the dominant species (Figure 8) that corresponds to a

zooplankton specific richness slight decrease (Figure 7). Copepod succession show the large dominance of calanoids (*T. longicornis*, *Acartia clausii*, *Centropages*, *Paracalanus* and *Pseudocalanus*) from January to July, harpacticoids relative abundance increasing in August to represent 50% of the zooplankton community in September (Figure 9).

There is no clear inter-annual trend in overall zooplankton community abundance (Table 3). However, meroplankton abundance shows a significant decreasing trend over time. Among copepods, the genera *Centropages* and *Paracalanus* show

significant increasing trends, while *Pseudocalanus* shows a significant decreasing trend. Mean inter-annual zooplankton abundances demonstrate some variability, with copepod species remaining the dominant group (Figure 10).




Diversity of both zooplankton and phytoplankton communities follow similar seasonal patterns, with the lowest diversity in April and May during the *P. globosa* bloom (Figure 7). However, there is a notable difference in February and March when phytoplankton diversity increases while zooplankton diversity decreases.


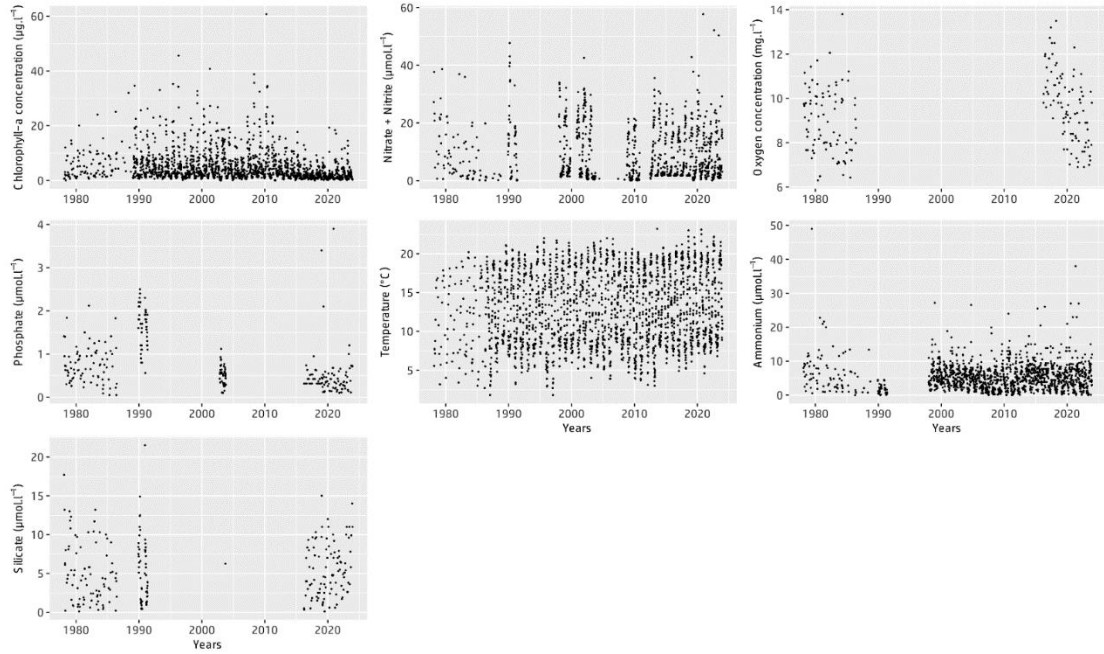

**Figure 2: Times series of the different physico-chemical parameters and chlorophyll-*a* concentrations measured from 1978 to 2023 as part of the IGA Gravelines monitoring program at the "Canal d'amenée" sampling station.**




**Table 3: Statistical summary (minimum, first and third quantiles, mean, median, maximum, length of data series) for main zooplankton and phytoplankton taxonomic groups and physico-chemical variables collected within the IGA monitoring program (1978-2023) at the "Canal d'amenée" station. The "Trend" column indicates whether there is an increasing or decreasing monotonic trend (1: single orange (green) arrow for increasing (decreasing) monotonic trend), a non-monotonic trend (2: two arrows indicating a shift in the time series), or no significant trend (grey arrow). The "Trend%unit/year" column provides the quantification of significant trends (as a percentage change per year using the Theil-Sen slope method).**

| Taxonomic group | Rank | Trend | Trend %unit/year | Min. | 1st Q. | Median | Mean | 3rd Q. | Max. | N |
|---|---|---|---|---|---|---|---|---|---|---|
| **Zooplankton (ind.m⁻³)** | | | | | | | | | | |
| *Appendicularia* | Class | ▲ | 0.04 | 0.04 | 9.47 | 40.49 | 206.76 | 134.43 | 7,066.78 | 408 |
| *Oikopleura (Vexillaria) dioica* | Species | ▲ | 0.04 | 0.04 | 9.25 | 39.75 | 188.30 | 131.05 | 7,051.24 | 406 |
| *Branchiopoda* | Class | na | | 0.10 | 2.92 | 14.70 | 64.61 | 50.39 | 1,058.10 | 98 |
| *Ctenophora* | Phylum | na | | 0.03 | 0.53 | 1.45 | 9.94 | 5.18 | 435.40 | 212 |
| *Chaetognatha* | Phylum | na | | 0.03 | 1.14 | 8.01 | 27.58 | 32.97 | 326.80 | 278 |
| *Copepoda* | Class | na | | 2.00 | 358.20 | 1,002.20 | 2,360.90 | 2,804.40 | 30,601.70 | 434 |
| *Temora* | Genus | na | | 0.26 | 75.62 | 266.00 | 1,009.12 | 909.96 | 16,711.11 | 429 |
| *Acartia* | Genus | na | | 1.18 | 48.98 | 204.07 | 648.86 | 702.24 | 10,050.60 | 426 |
| *Centropages* | Genus | ▲ | 0.06 | 0.10 | 15.79 | 56.50 | 267.15 | 175.09 | 7,729.28 | 415 |
| *Euterpina* | Genus | na | | 0.10 | 13.93 | 59.70 | 324.23 | 298.80 | 8,963.59 | 360 |
| *Paracalanus* | Genus | ▲ | 0.09 | 0.10 | 9.29 | 25.89 | 90.70 | 80.60 | 1,447.22 | 361 |
| *Pseudocalanus* | Genus | ▼ | -0.04 | 0.10 | 6.71 | 21.57 | 79.96 | 75.38 | 1,197.38 | 324 |
| Meroplankton total | Guild | ▼ | -0.01 | 0.01 | 24.53 | 156.25 | 428.26 | 447.91 | 11,818.61 | 486 |
| **Phytoplankton (cell.l⁻¹)** | | | | | | | | | | |
| *Bacillariophyceae* | Class | ▲ | 0.04 | 500 | 30,680 | 104,677 | 212,571 | 270,114 | 3,373,809 | 1,695 |
| *Chaetoceros* | Genus | ▲ | 0.04 | 10 | 1,900 | 8,700 | 68,557 | 36,726 | 5,340,000 | 1,387 |
| *Pseudo-nitzschia* | Genus | ▲ | 0.08 | 66 | 1,200 | 5,600 | 47,237 | 27,244 | 2,753,000 | 1,203 |
| *Paralia* | Genus | na | | 100 | 2,200 | 5,262 | 8,738 | 11,401 | 126,288 | 1,099 |
| *Rhizosolenia* | Genus | na | | 100 | 1,500 | 7,350 | 46,547 | 41,513 | 1,413,110 | 1,492 |
| *Skeletonema* | Genus | ▲ | 0.04 | 100 | 2,100 | 5,262 | 27,818 | 18,250 | 2,249,100 | 811 |
| *Thlassiosira+Porosira* | Genus | ▲ | 0.03 | 30 | 600 | 2,100 | 9,377 | 7,016 | 432,090 | 1,439 |
| *Leptocylindrus+Tenuicylindrus* | Genus | ▲ | 0.05 | 100 | 2,600 | 9,525 | 55,980 | 36,960 | 3,181,350 | 816 |
| *Asterionella+Asterionellopsis* | Genus | ▲ | 0.03 | 100 | 1,400 | 4,200 | 16,758 | 14,032 | 1,069,950 | 694 |
| *Eucampia+Climacodium* | Genus | ▼ | -0.02 | 80 | 700 | 2,100 | 11,107 | 7,895 | 809,900 | 852 |
| *Cryptophyceae* | Class | ▲ | 0.19 | 100 | 2,100 | 7,016 | 21,479 | 20,170 | 349,600 | 803 |
| *Dictyochophyceae* | Class | na | | 100 | 100 | 200 | 569.2 | 877 | 7,016 | 281 |
| *Dinophyceae* | Class | ▲ | 0.11 | 40 | 625 | 2,700 | 7,304 | 7,993 | 371,193 | 1,454 |
| *Prorocentrum* | Genus | ▲ | 0.11 | 0.2 | 200 | 877 | 3,715 | 3,508 | 87,700 | 507 |
| *Phaeocystis globosa* | Species | na | | 3 | 125.655 | 1,441.448 | 5,180,501 | 5,793,562 | 57,734,729 | 290 |
| *Dinophyceae/Bacillariophyceae* | Ratio | ▲ | 0.05 | 0.00019 | 0.00844 | 0.02077 | 0.05556 | 0.05195 | 2.49480 | 1,424 |
| **Physico-chemical** | | | | | | | | | | |
| Temperature (°C) | | ▲ | 0.003 | 1.80 | 8.90 | 12.60 | 12.95 | 17.30 | 23.20 | 2,292 |
| Salinity | | na | | 31.40 | 33.60 | 33.90 | 33.87 | 34.20 | 37.16 | 913 |
| NH4 (µmol.l⁻¹) | | na | | 0.03 | 2.42 | 4.79 | 5.32 | 7.21 | 49.00 | 1,509 |
| NO2 (µmol.l⁻¹) | | na | | 0.01 | 0.23 | 0.40 | 0.51 | 0.65 | 16.00 | 902 |
| NO3 (µmol.l⁻¹) | | na | | 0.09 | 1.61 | 4.40 | 8.73 | 14.90 | 57.00 | 1,310 |
| NO2+NO3 (µmol.l⁻¹) | | na | | 0.09 | 1.87 | 4.69 | 9.02 | 15.16 | 57.70 | 1,099 |
| PO4 (µmol.l⁻¹) | | ▼ | -0.01 | 0.05 | 0.32 | 0.58 | 0.79 | 1.05 | 1.05 | 289 |
| SiOH (µmol.l⁻¹) | | na | | 0.1 | 2.00 | 4.55 | 5.27 | 8.01 | 21.51 | 242 |
| Chlorophyll-*a* (µg.l⁻¹) | | ▼ | -0.02 | 0.01 | 1.38 | 2.85 | 4.90 | 6.26 | 60.76 | 1,901 |





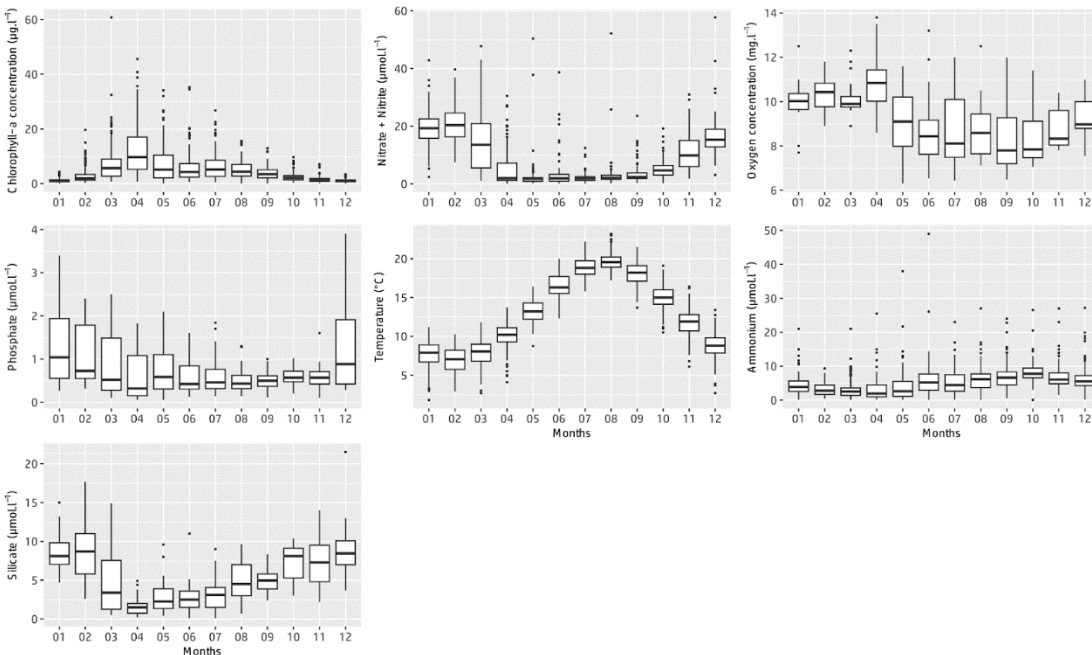

**Figure 3: Monthly box-and-whisker plots of the main physico-chemical parameters at the "Canal d'amenée" sampling station as part of the IGA Gravelines monitoring program over the period 1978–2023.**

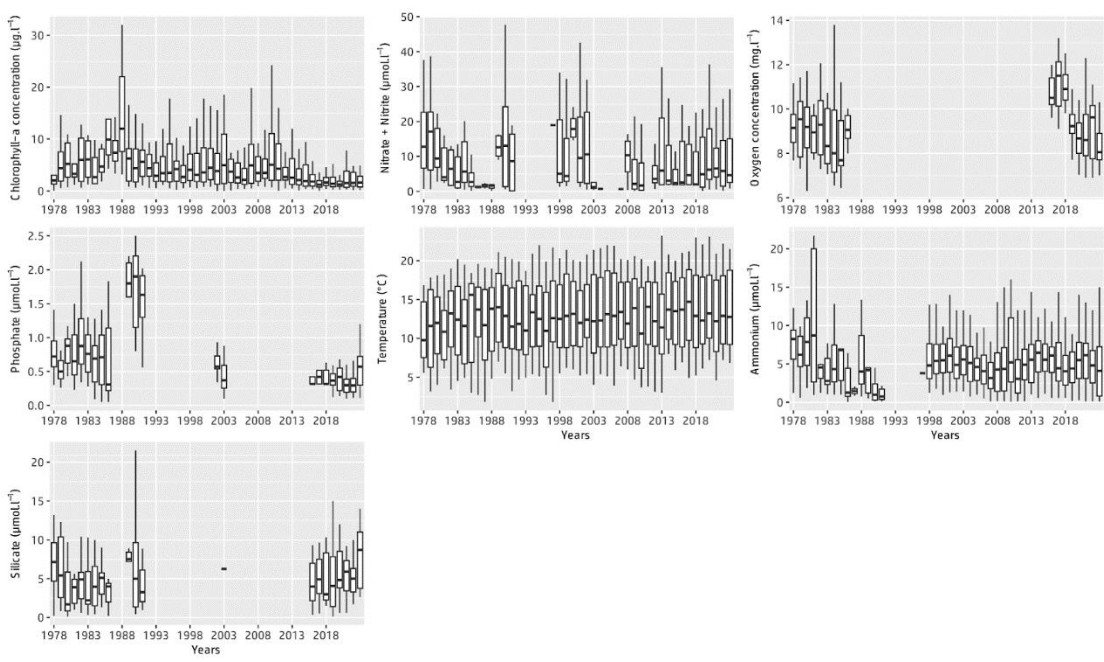


**Figure 4: Annual box-and-whisker plots of the main physico-chemical parameters at the "Canal d'amenée" sampling station as part of the IGA Gravelines monitoring program over the period 1978–2023. For improved data visualization, outliers are not represented.**



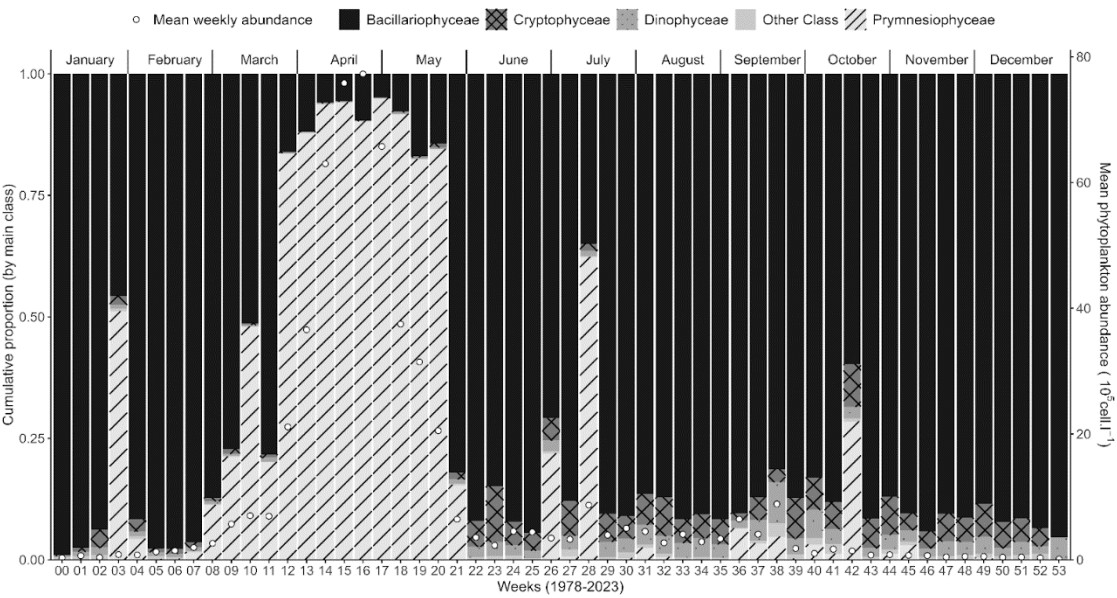

**Figure 5: Weekly average variability (1978-2023) of the major phytoplankton groups (Prymnesiophyceae mainly *Phaeocystis globosa*, Bacillariophyceae, Dinophyceae, Cryptophyceae, and other phytoplankton) at the "Canal d'amenée" sampling station, as part of the IGA Gravelines monitoring program. The vertical bars illustrate the relative abundances of these groups (%), while the black circles indicate the mean weekly total phytoplankton abundance ($10^5$ cells.l$^{-1}$).**

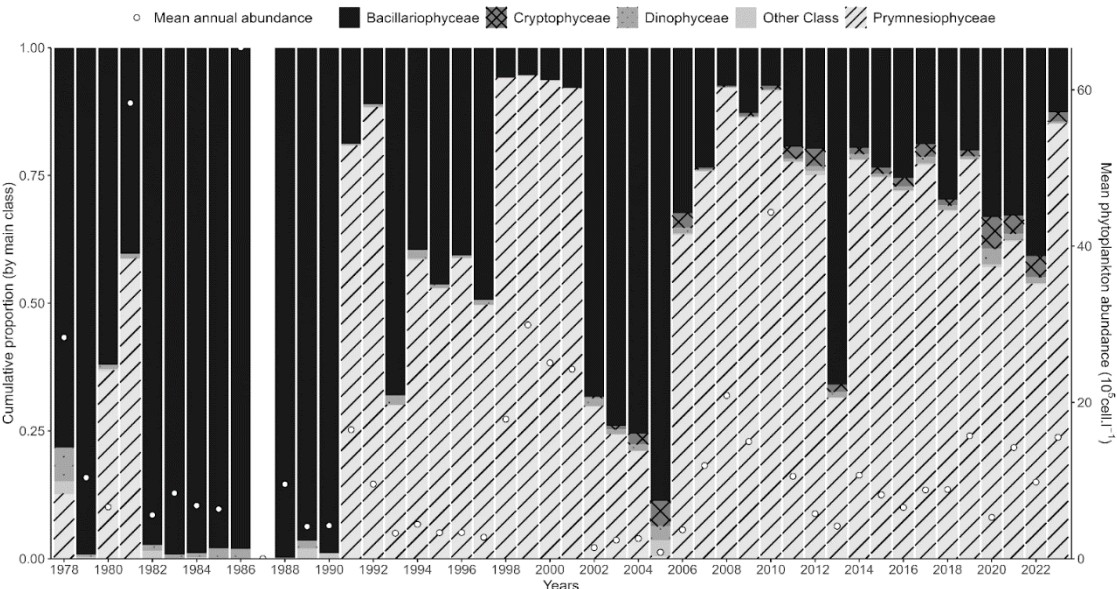

**Figure 6: Inter-annual variability of the major phytoplankton groups (Prymnesiophyceae mainly *Phaeocystis globosa*, Bacillariophyceae, Dinophyceae, Cryptophyceae, and other phytoplankton) at the "Canal d'amenée" sampling station, as part of the IGA Gravelines monitoring program. The vertical bars represent the relative abundances of these groups (%), while the black circles indicate the mean annual total phytoplankton abundance ($10^5$ cells.l$^{-1}$).**





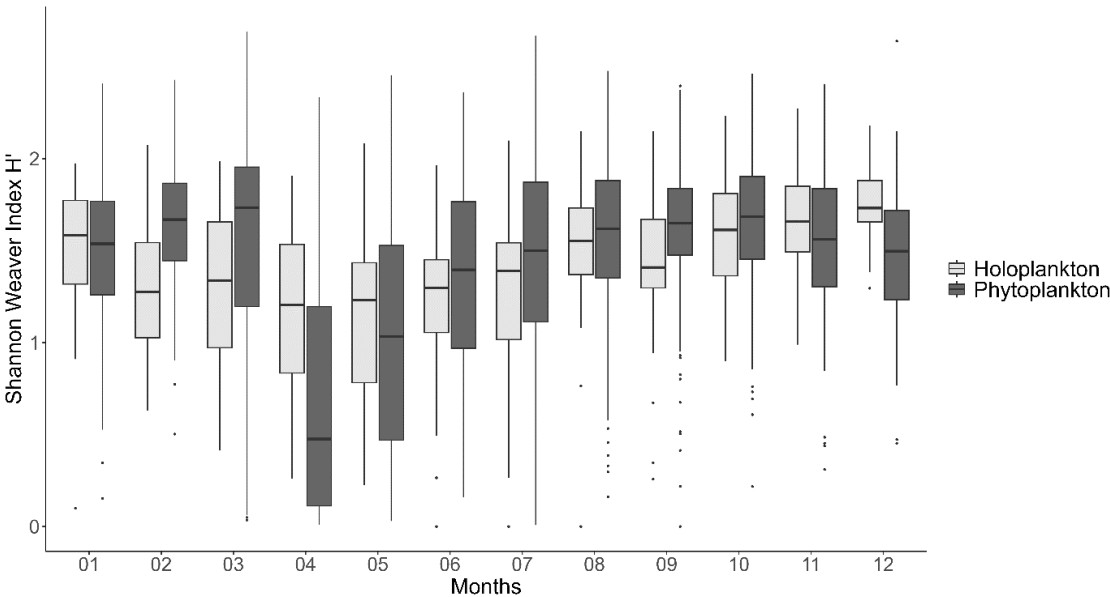

**Figure 7: Mean seasonal variation (monthly scale, 1978-2023) of the Shannon-Weaver diversity index for both phytoplankton and holo(zoo)plankton at the "Canal d'amenée" sampling station, as part of the IGA Gravelines monitoring program.**

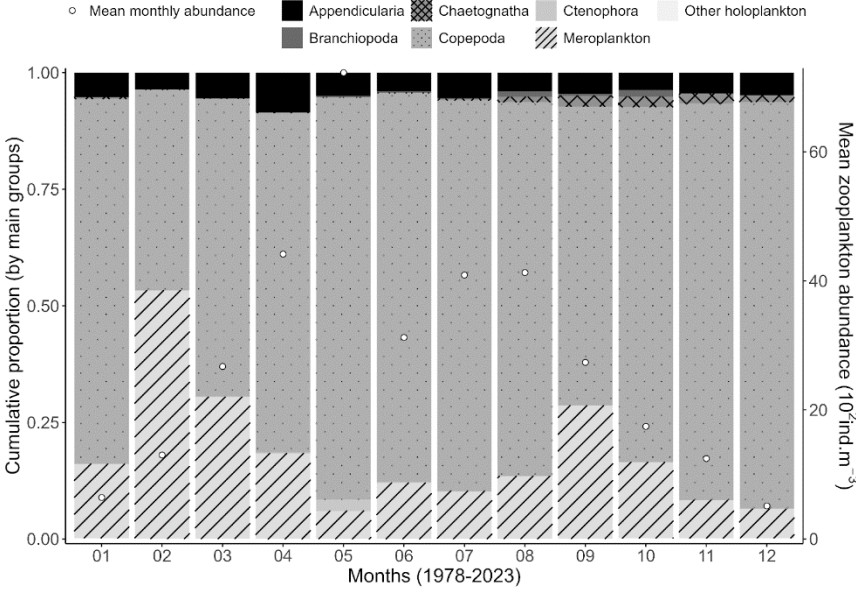

**Figure 8: Mean seasonal variability (1978-2023) of the major zooplankton groups (Appendicularia, Branchiopoda, Copepoda, Chaetognatha, Ctenophora, Meroplankton, and other holoplankton) at the "Canal d'amenée" sampling station of the IGA Gravelines monitoring program. The vertical bars indicate the relative abundances of these groups (%), while the black circles represent the mean monthly total abundance ($10^2$ ind.m$^{-3}$).**

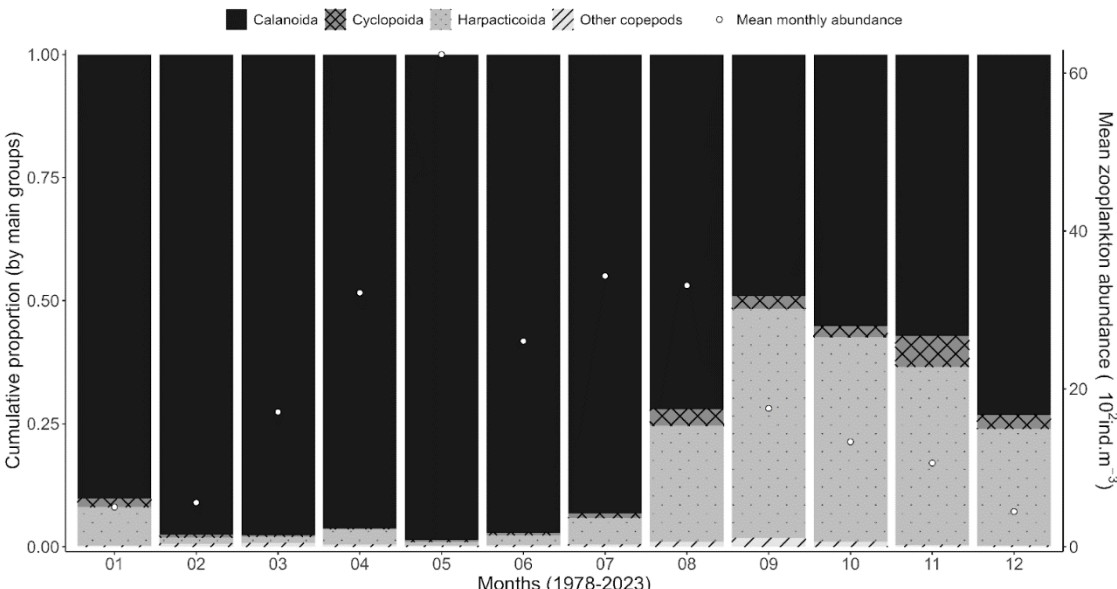

**Figure 9: Mean monthly variability (1978-2023) of copepods orders (Calanoida, Cyclopoida and Harpacticoida) at the "Canal d'amenée" sampling station of the IGA Gravelines monitoring program. The vertical bars illustrate the relative abundances of these groups (%), while the black circles show the mean monthly total abundance ($10^2$ ind.m$^{-3}$).**

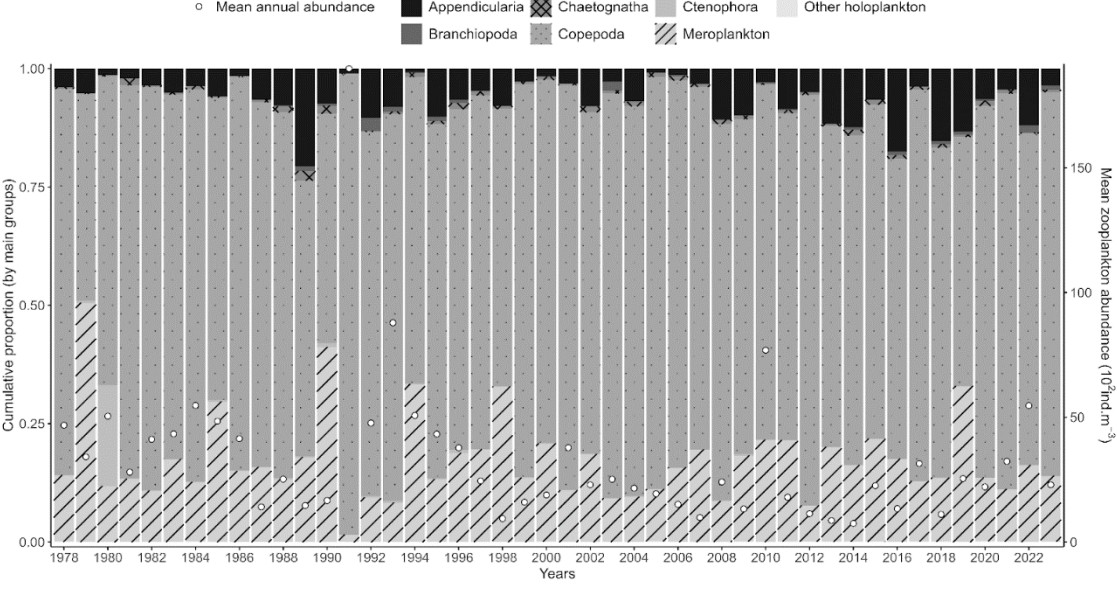

**Figure 10: Inter-annual variability of the major zooplankton groups (Appendicularia, Branchiopoda, Copepoda, Chaetognatha, Ctenophora, Meroplankton, and other holoplankton) at the "Canal d'amenée" sampling station of the IGA Gravelines monitoring program. The vertical bars indicate the relative abundances of these groups (%), while the black circles represent the mean annual total abundance ($10^2$ ind.m$^{-3}$).**





## 8    Discussion and conclusion

The IGA Gravelines "Canal d'amenée" data series, which began in 1978, represents the longest physico-chemical and
planktonic data series in the southern bight of the North Sea. This time series captures the dynamics of phytoplankton, mainly
dominated by Bacillariophyceae and the Prymnesiophycae, *Phaeocystis globosa*, as well as zooplankton species, with a focus
on copepods. The dataset reveals temporal successions and multi-year trends in plankton communities, including annual
phenology characterized by an initial increase in Bacillariophyceae abundance, followed by the early spring bloom of *P.
globosa*, which coincides with high abundances of the potential harmful alga *Pseudo-nitschzia* sp.

The Bacillariophyceae/*P. globosa* bloom is one of the most structuring event in the planktonic community of this region of the
Eastern English Channel (Schapira et al., 2008, Grattepanche et al., 2011), influencing not only plankton but also bacteria
(Lamy et al., 2006) and benthic communities (Dauvin et al., 2008; Denis and Desroy, 2008; Spilmont et al., 2009). The IGA
survey shows that this bloom varies significantly in intensity and relative proportion between years. During this bloom, annual
chlorophyll-*a* concentrations peak, while nutrients levels hardly decrease. This period also sees a significant drop in the annual
proportion of copepod species (the most abundant primary consumers in the zooplankton community) showing the impact of
this bloom on the entire pelagic ecosystem, as previously described in this region (Lancelot et al., 2005).

The inter-annual variability in plankton dynamics was concomitant with a significant increasing trend in seawater temperature
and a significant decreasing trend in phosphate concentrations. Both of these parameters were already described as drivers of
*P. globosa* blooms (Hernández Fariñas et al., 2015, Karasiewicz et al., 2018). Given the variety of parameters measured, the
frequency of measurements, and the length of the dataset, this series is exceptionally well suited for studying the dynamics of
planktonic communities and the effects of anthropogenic pressures on them.

Indeed, similar long-term data series have proven to be valuable resources for scientific research, leading to a significant
number of publications across various topics. For instance, the "Point B" series from the Laboratoire d'Océanographie de
Villefranche sur Mer (LOV) in the Mediterranean Sea (Romagnan et al., 2015; Romagnan et al., 2016; Feuilloley et al., 2022),
the "L4" series from the Plymouth Marine Laboratory in the Western Channel (Harris, 2010; Widdicombe et al., 2010; McEvoy
et al., 2023) and the "SRN" series from Ifremer's LER-BL laboratory in the Eastern Channel-North Sea (Lefebvre and
Devreker, 2023; Lefebvre et al., 2011) have generated a substantial number of scientific articles addressing a wide range of
topics. These include plankton dynamics (Vandromme et al., 2011; John et al., 2001; Lefebvre and Devreker, 2023), climate
change (Corona et al., 2024; Kapsenberg et al., 2017; Parravicini et al., 2015), harmful algal blooms (HABs) (Karasiewicz et
al., 2020; Karasiewicz and Lefebvre, 2022), food webs (Atkinson et al., 2015), and the carbon cycle (González-Benítez et al.,
2019). Such extensive, long term, multi-parameter datasets can also be critical for assessing environmental quality within the
framework of European Directives (such as the Water Framework Directive WFD or the Marine Strategy Framework Directive
MSFD) and Regional Marine Policies (such as the OSPAR or Barcelona Conventions) (McQuatters-Gollop et al., 2019;
Lefebvre and Devreker, 2020).



The IGA Gravelines "Canal d'amenée" series has been used in various studies, including analyses of long-term temperature fluctuations and their link with the North Atlantic Oscillation (NAO) (Woehrlings et al., 2005). The dataset, covering measurements from 1975 to 1992, has been used to investigate trophic relationships (Le Fevre-Lehoerff et al., 1993) and cycles within the context of climatic changes (Le Fevre-Lehoerff et al., 1995), incorporating data on water temperature, salinity, suspended matter, nutrients, chlorophyll-*a*, and zooplankton. Halsband-Lenk and Antajan (2010) showed the utility of this

time series fir defining regional multi-metric food web indices using temperature, salinity, chlorophyll pigments, phytoplankton and zooplankton abundance data. The series has also been identified as a series-of-interest by the ICES WGZE (Working Group on Zooplankton Ecology) and has been integrated into NOAA's METABASE (https://www.st.nmfs.noaa.gov/copepod/time-series/fr-30101/). Despite the highest scientific value of this dataset, the number of publications focused on plankton and hydrology remains relatively limited compared to other datasets, such as "L4" and

"Point B" (Google Scholar results for "plankton" and the station name since 2020: 128 for "L4" and 69 for "Point B"). Publications related to benthic fauna from the IGA Gravelines survey are more numerous.

Moreover, studying plankton dynamics using such data series can also provide critical insights into preventing cooling problems at Nuclear Power Plants. High biomass blooms can obstruct cooling systems by either physically blocking the flow or altering water viscosity, leading to reduced efficiency and potential operational issues. Such events has been documented

in literature, including cases involving gelatinous species at Gravelines (Antajan et al., 2014) and HABs globally (Wang et al., 2022). At the Gravelines NPP, in addition to issues caused by gelatinous blooms, particularly from *Pleurobrachia pileus*, blooms of *Phaeocystis globosa* can also affect cooling systems. Consequently, it is crucial to understand the drivers of *P. globosa* blooms, including the parameters that influence their intensity and timing. Developing early warning systems to anticipate such blooms and implementing preventive or mitigative measures is thus essential for maintaining operational

efficiency and avoiding disruptions.

The data acquired through the IGA program follow the FAIR principles, which define a set of rules to facilitate Findability, Accessibility, Interoperability and Reusability of data and associated metadata. This dataset from Gravelines is particularly well suited for collaborative annotation work, which could enhance forecasting capabilities. This process involves identifying and labelling recurring, rare, and extreme events in the time series by experts. Recent advancements in machine learning have

introduced several tools to assist with pattern recognition and automatic segmentation of time series data. Techniques include fixed-length window segmentation (Van Hoan et al., 2017), sliding window approaches using autoencoders (Längkvist et al., 2014), Expectation-Maximization models (Poisson-Caillault and Lefebvre, 2017), Hidden Markov Models (Dias et al., 2015; Rousseeuw et al., 2015), and Multi-Level Spectral Clustering (Grassi et al., 2020). These methods can be used to isolate patterns in the time series, which can then be annotated with relevant event labels based on the expertise of the annotators.

This labelled database could be used as a reference training set by the scientific community, addressing specific needs in artificial intelligence. It would be particularly valuable for developing algorithms, calibrating models, and implementing digital prediction and warning systems. Such advancements could enhance biologists' understanding of marine dynamics, providing



new insights into plankton community functioning and environmental states evolution, including potential trends or regime shifts in the context of global environmental change.

The recent decision by EDF to provide open access to the IGA Gravelines "Canal d'amenée" monitoring data offers significant opportunities for scientific advancement. This will facilitate the utilization of the time series in ways similar to other long-term datasets. By detailing the characteristics of the Gravelines "Canal d'amenée" series, this article provides the information needed for future scientific investigations and management applications to researchers and managers, and thereby contributing to a deeper understanding of coastal ecosystems in the Channel-North Sea.

**Data and code availability**

The IGA-HP Gravelines dataset is publicly available via https://doi.org/10.17882/102656 (Hydrology and Plankton monitoring programme at the Gravelines coastal station, 2024).

The R-package **TTAinterfaceTrendAnalysis** is available on the CRAN website (Comprehensive R Archive Network - https://cran.r-project.org/package=TTAinterfaceTrendAnalysis/index.html ; last access: January 23, 2024).

**Author contribution**

DD, GW, AL wrote the paper. DD created all the figures and tables. GW coordinates the ecological monitoring of the Gravelines NPP for the IGA program. AL led the conceptualization and overall writing of the paper.

**Competing interests**

The authors declare that they have no conflict of interest.

**Acknowledgments**

The authors wish to express their gratitude to the national managers of the IGA program: Thillaye Du Boullay Hervé, Drévès Luc and Ropert Michel, the coordinators of the Gravelines Nuclear Power Plant: Le Fèvre-Lehoërff Geneviève, Lefebvre Alain, Antajan Elvire and Wacquet Guillaume, and all the technical teams and subcontractors who have been involved in sampling and analysing the various compartments followed within the frame of the monitoring program since 1978.

They also extend their thanks to EDF for providing access to the data and for reviewing the article.





**Financial support**

The data were collected as part of the IGA (Impact des Grands Aménagements) monitoring program (market agreement
n°C3499C0490) conducted by IFREMER with financial support from EDF (Électricité de France). These data remain the
property of EDF and are made accessible to the scientific community for research purposes.

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
