# Peer review of "A 45-year hydrological and planktonic time series in the South Bight of the North Sea"

_Earth System Science Data, 2024_

## Author Response (AR1)

**RC1**

It was a pleasure to read this paper, and to see this incredibly valuable dataset be published and accessible. I have almost no issues with the text as is, other than a small comment for page1 - just after line 25. the authors say that long term plankton time series are not available - but there are some very important ones for the North Sea - and summarised in Holland et al., Holland, Matthew M., et al. "Major declines in NE Atlantic plankton contrast with more stable populations in the rapidly warming North Sea." Science of the Total Environment 898 (2023): 165505 and also summarised in Holland, Matthew M., Angus Atkinson, Mike Best, Eileen Bresnan, Michelle Devlin, Eric Goberville, Pierre Hélaouët et al. "Predictors of long-term variability in NE Atlantic plankton communities." Science of the Total Environment 952 (2024): 17579. There are also descriptions of the plankton data in the recent OSPAR biodiversity report under Quality Status Reporting.  But other than that small fact, - its a well written paper, and a dataset that will be invaluable for our long term understanding of WQ and plankton change.

We sincerely thank the reviewer for their positive comments and interesting suggestions on our manuscript. We are pleased that the importance of this dataset has been recognized and that it is considered as a valuable contribution to long-term understanding of plankton dynamics and water quality changes.

Regarding the specific comment on long-term plankton time series, we appreciate that the reviewer highlights the important datasets for the North Sea, as summarized in the referenced papers by Holland et al. (2023, 2024) and the recent OSPAR Biodiversity Report.

We acknowledge that we have explicitly mentioned the extensive spatio-temporal datasets from the well-known Continuous Plankton Recorder (CPR) survey or the long-term North Sea datasets submitted to the OSPAR COBAM group. Regarding CPR data, our focus was primarily on Eulerian surveys, while the CPR is inherently Lagrangian in nature. In addition, some other valuable datasets were not included due to accessibility issues, as they are not always readily available: only datasets from France, Sweden (e.g., from the Kattegat and Skagerrak regions, which are at the interface between the North Sea and the Baltic Sea), and the UK have associated DOIs, facilitating easier access and citation.

We will add these references to the article to clarify our statement about the availability of long-term plankton time series.

**RC2**

**General Comments**

The authors present a long-term dataset of hydrological and planktonic observations in the South Bight of the North Sea.

These data are particularly precious and relevant as their combination represents an incredible volume of open access monitoring data. Moreover, these observations offer significant opportunities for scientific advancement, being potentially beneficial for other ecological and management applications. Hence, this study certainly meets the ESSD criteria for data availability.

The presentation of the study site, monitoring strategy and methods is well written and very detailed even if more details could be provided (as detailed below).

We sincerely thank the reviewer for their positive comments and interesting inputs on our manuscript. We are pleased that the importance of this dataset has been recognized and that it is considered as a valuable contribution to long-term understanding of plankton dynamics and water quality changes.

**Specific Scientific Comments**

As regards data quality no reference is made to sensors' maintenance and calibration and possible inter-calibration of instruments as regards temperature, salinity, turbidity and oxygen concentration. For example, no reference is made to collection of discrete samples to periodically calibrate the dissolved oxygen data recorded by the sensor, as initially done with the Winkler method. The in-situ samples are needed to fulfill quality assurance standards and correct any potential bias and drift in the sensors.

Unfortunately we do not have supplementary details regarding the sensor calibration procedures done by the operators.

For oxygen concentration, the Winkler method was used prior to the 1990's. Thereafter, sensor measurements were done by COFRAC accredited laboratories following NF EN ISO standards, as was the case for other sensor-measured parameters. In addition, the quality control in the Quardige² database (detailed in the §5) ensures that no strong inconstancy appears in the data series. Accreditation, standards and data validation processes act as a safeguard against questionable data.

**Technical Comments**

Line 75: "This location…" instead of "This situation…"

Thank you for this suggestion. The change has been made.

Line 94: "The water was analysed…" instead of "The water is analysed …"

Thank you for this suggestion. The change has been made.

Figure 4 appears in the main text before Figure 3 (line 268 vs line 271). Reverse the order of the figures in the manuscript to be consistent.

Thank you for pointing out this error. The correct order has been restored.

Line 293: "peak value..." instead of "pic value....".

Thank you for pointing out this error. The correction has been made.

Line 294: Please check month (March or April?) and related zooplankton mean monthly abundance.

Thank you for out pointing this error. The maximum abundance of zooplankton was effectively observed in May (and not in March). The correction has been made.

Line 324: Any two arrows or grey arrow is reported in Table 3. Please delete "…. A non-monotonic trend (2: two arrows indicating a shift in the time series), or no significant trend (grey arrow)…"

Thank you for this remark. Table caption has been corrected.

Line 400: "the utility of this time series for…" instead of "the utility of this time series fir…"

Thank you for this correction. The change has been made.

**RC3**

Devreker et al. presented a highly valuable in situ dataset spanning 45 years from a coastal station. The article is well-written and clearly presented. Such initiatives and open datasets are crucial for advancing ecological studies, as they provide a broader understanding of phytoplankton dynamics over the long term. I have some questions and minor corrections, detailed below:

We sincerely thank the reviewer for their positive comments and interesting inputs on our manuscript. We are pleased that the importance of this dataset has been recognized and that it is considered as a valuable contribution to long-term understanding of plankton dynamics and water quality changes.

HAB events are expected to increase in the coming decades. It would be interesting to evaluate whether there is an observed increase in the frequency of occurrences of potentially toxic phytoplankton species in large concentrations across your 45 years of data.

We agree with this comment and will specifically address this point in a future ecological manuscript based on this (and other) datasets.

Line 28: "scarce" instead of "scare"

Thank you for pointing out this error. The correction has been made.

Line 29: Reword as "and are responsible for approximately 50% of global primary production" instead of "contribute to 50% of the Earth's chlorophyll biomass."

Thank you for this suggestion. The change has been made.

Line 60: The acronym "HAB" is not defined upon its first use

Thank you for pointing out this lack. The definition has been added.

Line 94: what is an "oceanographic bucket"? I suggest deleting "using an oceanographic bucket" and replace "The water is analyzed" by "water samples are analyzed"

Thank you for this suggestion. The change has been made.

Line 95: "at every sampling" instead of "at every instance".

Thank you for this suggestion. The change has been made.

Section 3.2.1: Are corrections performed to account for possible biases between different sensors? Are the sensors calibrated regularly?

Unfortunately we do not have supplementary details regarding the sensor calibration procedures done by the operators.

Sensor measurements were done by COFRAC accredited laboratories following NF EN ISO standards, as was the case for other sensor-measured parameters. In addition, the quality control in the Quardige² database (detailed in the §5) ensures that no strong inconstancy appears in the data series. Accreditation, standards and data validation processes act as a safeguard against questionable data.

Section 3.3.1: Are micro-, nano-, and pico-plankton identified during sampling? If not, please specify which size classes are considered.

Phytoplankton cell counts are performed using inverted microscopy using a 20X Plan Ph1 0.5NA objective. For this reason, only one size group in the phytoplankton population (micro-plankton) are identified. Details have been added in 3.3.1 (and 3.3.2 for zooplankton).

Line 154: The Shannon-Weaver Index is a diversity, not a richness, index. Replace "Species richness calculations" with "Phytoplankton species diversity calculations."

Thank you for this suggestion. The change has been made.

Equation 1: explain parameters used in the equation. There is an error in the H' formula, it should be:

 -sum (pi ln(pi))

Thank you for pointing out this error. The equation has been corrected.

Section 3.3.2: How is zooplankton sampling performed? Is it based on time intervals? Is the volume of water passing through the net quantified?

Each sample is collected one month apart, typically mid-month, though adverse weather conditions can cause variations of several days.

During sampling, the net is maintained at a depth of 2 to 5 meters below the surface. This depth is chosen due to the currents and turbulence, which homogenize the water column.

The sampled volume is measured using a flow meter attached to the WP2 net opening. Sampling intervals are adjusted depending on the target (10 minutes for gelatinous zooplankton, and 3 minutes for zooplankton abundances).

This information has been added to the Materials and Methods section.

Line 217: Avoid leaving any cells empty, as this can cause issues for users working with software like MATLAB. Instead, replace empty cells with "NaN" or use a numerical code for missing data, such as "-9999."

We apologize for the inconvenience caused by the empty cells being incompatible with MATLAB data import. All calculations and data formatting were performed using the R programming language, where empty cells did not cause any issues. While we understand the importance of ensuring compatibility across platforms, the data has already been submitted to SEANOE, and we are unable to make modifications to its encoding at this stage.

Line 241: "searching for data that might be"…

Thank you for this suggestion. The change has been made.

Line 269: Is the trend in phytoplankton diversity consistent with the observed trends in chlorophyll concentration?

The overall trend in phytoplankton abundance seems to be increasing, while the chlorophyll trend is decreasing (a phenomenon well-documented in the English Channel, particularly in the early 2000s). This could potentially be due to a modification in phytoplankton species composition and particularly in the ratio between pico-nano-phytoplankton versus microphytoplankton. This change in the size fraction of phytoplankton has already been documented in the scientific literature (Louchart et al. 2020; Lefebvre et al. 2019) using flow cytometry data. However, the limitation of cell size detection due to the use of optical microscope didn't allow to observe pico and nanophytoplankton in the IGA monitoring network. This observation warrants further investigation and could be explored in greater detail in a dedicated environmental manuscript based on this (and other) datasets.

Louchart Arnaud, Lizon Fabrice, Lefebvre Alain, Didry Morgane, Schmitt François G., Artigas Luis Felipe (2020). Phytoplankton distribution from Western to Central English Channel, revealed by automated flow cytometry during the summer-fall transition . Continental Shelf Research , 195, 104056 (16p.) . Publisher's official version: https://doi.org/10.1016/j.csr.2020.104056 , Open Access version : https://archimer.ifremer.fr/doc/00601/71324/

Lefebvre Alain, Poisson-Caillault Emilie (2019). High resolution overview of phytoplankton spectral groups and hydrological conditions in the eastern English Channel using unsupervised clustering. Marine Ecology Progress Series , 608, 73-92 . Publisher's official version: https://doi.org/10.3354/meps12781 , Open Access version : https://archimer.ifremer.fr/doc/00479/59043/